# Effects of Drying Methods on the Physicochemical Aspects and Volatile Compounds of *Lyophyllum decastes*

**DOI:** 10.3390/foods11203249

**Published:** 2022-10-18

**Authors:** Bin Yang, Jianhang Huang, Wensong Jin, Shujing Sun, Kaihui Hu, Jiahuan Li

**Affiliations:** 1College of Life Sciences, Fujian Agriculture and Forestry University, Fuzhou 350002, China; 2Edible Fungal Research Institute (Gutian), Fujian Agriculture and Forestry University, Ningde 352200, China

**Keywords:** *Lyophyllum decastes*, drying methods, physicochemical aspects, volatile compounds, gas chromatography-ion mobility spectrometry (GC-IMS), principal component analysis (PCA)

## Abstract

In this study, fresh *Lyophyllum decastes* was dried using hot air drying (HAD), hot air combined with vacuum drying (HAVD), and vacuum freeze drying (VFD). Additionally, the quality and volatile compounds were analyzed. VFD achieved the best color retention, the highest rehydration capacity, and the slightest damaged tissue structure; however, it recorded the longest drying time and the highest energy consumption. HAD was the most energy-efficient of the three methods. Furthermore, the products with more hardness and elasticity were obtained by HAD and HAVD—this finding was convenient for transportation. In addition, GC-IMS demonstrated that the flavor components had significantly changed after drying. A total of 57 volatile flavor compounds was identified, and the aldehyde, alcohol, and ketone compounds were the primary ingredient of the *L. decastes* flavor component, whereby the relative content of the HAD sample was apparently higher than HAVD and VFD. Taken together, VFD was better at preserving the color and shape of fresh *L. decastes*, but HAD was more appropriate for drying *L. decastes* because of the lower energy consumption, and was more economical. Meanwhile, HAD could be used to produce a more intense aroma.

## 1. Introduction

*Lyophyllum decastes* is accepted by general consumers, owing to its delicious taste, special flavor, and abundant nutrients. Furthermore, its fruiting body has been reported to contain a wide array of phytochemicals which are claimed to exert many health benefits, including antioxidants [1], antidiabetics [2], hypolipidemics [3,4], immunoregulation [5], and antitumor properties [6]. With the development of industrialized cultivation techniques, the production of *L. decastes* has increased substantially in recent years. According to the data released by the Chinese Edible Mushroom Association, the production of *L. decastes* in 2018 increased by 257% year-on-year compared with 2017. However, the fresh *L. decastes* is highly perishable due to its high moisture content (around 89% wet basis), which has easily led to the quality deterioration and great economic losses for the *L. decastes* industry.

Dehydration, as a convenient and effective process in the food industry, can reduce moisture to a safe level to prolong the shelf life [7]. However, the different drying methods have significantly affected the characteristics of dried products, such as the physicochemical properties, flavor characteristics, drying characteristics, and texture properties [8,9]. Hot air drying (HAD), vacuum drying (VD), and vacuum freeze drying (VFD) are the three common drying process for mushrooms. Additionally, different methods have various advantages during the drying process of different types of edible fungi. HAD could be used as an appropriate drying method, rather than others, for shiitake mushrooms to produce a typical aroma [10]. VD is usually combined with other drying methods, such as hot air combined with vacuum drying (HAVD) and microwave-vacuum drying (MWVD), which can effectively reduce the loss of active constituents and nutrients, improve color retention, and increase rehydration in *Lentinula edodes* [11,12]. However, VFD is more preferable for *Pleurotus ostreatus* preservation than VD in terms of color, size reduction, rehydration properties, and active constituent content [13]. Therefore, it is vital to find an appropriate drying technology for different varieties of edible fungi in the post-harvest preservation. However, as far as we know, few studies are available on the drying of *L. decastes.*

The flavor of mushrooms is one of the driving forces for their consumption, and it is affected by the drying methods. Gas chromatography ion migration spectrum technology (GC-IMS) with automatic headspace sampling (HS), as an emerging mass spectrometry technology, is gaining increasing popularity in the field of volatile flavor compound analysis [14]. In recent years, this technology has been successfully used to analyze the dynamic change in volatile compounds for mushrooms in the drying process and to distinguish the differences in flavor characteristics of mushrooms with different drying method treatments [15,16]. Compared with traditional spectrometry technology, HS-GC-IMS combines the advantages of both GC and IMS, such as high sensitivity, high accuracy, high efficiency, and low detection limits, which not only can acquire intuitive 2D and 3D maps, but can also increase the overall peak capacity, resolution, and qualitative analysis accuracy of compounds, especially for the analysis of some isomers [14,17]. Until now, the effects of drying methods on the volatile flavor compounds of *L. decastes* have not been subjected to research.

Therefore, the objective of this work was to explore the effects of three drying methods (HAD, HAVD, and VFD) on drying characteristics, physicochemical properties (color, rehydration), texture properties (hardness, elasticity, chewiness, and microstructure), and flavor characteristics (volatile compounds) of *L. decastes*. The result may present a theoretical reference for selecting a drying technology to achieve high quality, high efficiency, low energy consumption, and facilitate the industrial production of dried *L. decastes*.

## 2. Materials and Methods

### 2.1. Materials

Fresh *L. decastes* samples with similar size, maturity, and without mechanical damage were purchased from the Zhenjunzi Biotechnology Co., Ltd., in Ningde, Fujian Provience, China and were refrigerated at 4 °C (EBM190GVA, Electrolux (China) Electrical Co., Ltd., Shanghai, China) for no more than four days before the drying treatments. The initial moisture content of fresh *L. decastes* was 89.52 ± 0.16% wet basis (g/g, w.b.) measured by a moisture analyzer (XY-110MW, Chongzhou Lucky Electronic Equipment Co., Ltd., Chongzhou, China).

### 2.2. Drying Treatment

The drying parameters were obtained from the previous experiments in our laboratory. The fresh *L. decastes* samples were dried by hot air drying (HAD), hot air combined with vacuum drying (HAVD), and vacuum freeze drying (VFD), until the final moisture content was ≤10% wet basis (g/g, w.b.) according to national standards; then, the drying was finished.

#### 2.2.1. Hot Air Drying (HAD)

The fresh *L. decastes* sample (1 kg) was uniformly spread onto a tray, and then put into the drying room of an electro-thermostatic blast oven (DHG-9240A, Jinghong Experimental Equipment Co., Ltd., Shanghai, China). The drying parameters were: primary temperature at 42 °C, treated for 3 h, before rising to 55 °C until the moisture content of the sample was ≤10% (g/g, w.b.).

#### 2.2.2. Hot Air Combined with Vacuum Drying (HAVD)

Firstly, the fresh *L. decastes* sample (1 kg) was evenly spread onto a tray, and then placed into the drying chamber of the electro-thermostatic blast oven (DHG-9240A, Jinghong Experimental Equipment Co., Ltd., Shanghai, China) at 45 °C and treated for 4 h, before rising to 55 °C until the moisture content of sample was below 75% (g/g, w.b.); then, it was transferred to the vacuum drying oven (Weifang North Pharmaceutical Equipment Co., Ltd., Weifang, China.) at 60 °C, and the pressure of the drying chamber was set at −0.08 MPa, until the moisture content of the sample was ≤10% (g/g, w.b.); then, the drying was finished.

#### 2.2.3. Vacuum Freeze Drying (VFD)

The fresh *L. decastes* sample (1 kg) was pre-frozen in a −80 °C refrigerator (DW-86L626, Qingdao Haier Special Electrical Appliance Co., Ltd., Qingdao, China.) for 10 h, and then moved to a vacuum freeze dryer (LJG-12, Songyuan Huaxing Technology Development Co., Ltd., Beijing, China). The drying parameters were: degree of vacuum of < 5 Pa, cold-trap temperature of −45 °C until the moisture content of the sample was ≤10% (g/g, w.b.); then, the drying was finished.

### 2.3. Drying Time and Unit Energy Consumption (UEC)

#### 2.3.1. Drying Time 

The moisture content of samples treated with hot air drying, hot air combined with vacuum drying, and vacuum freeze drying was tested every 0.5 h by a moisture analyzer (XY-110MW, Chongzhou Lucky Electronic Equipment Co., Ltd., Chongzhou, China.) after 12 h, 10 h, and 16 h, respectively. When the moisture content of the samples was below 10% (g/g, w.b.), the drying was finished, and the drying time was recorded.

#### 2.3.2. Unit Energy Consumption (UEC)

The UEC in the drying process was calculated by Equation (1) according to the output power of equipment and drying time.
(1)$$\mathrm{UEC}=\frac{P\times T}{m}$$
where *P* represented the output power of equipment (kW/h), *T* was the total of drying time (h), and m represented the amount of fresh sample (kg).

### 2.4. Physicochemical Properties Analysis

#### 2.4.1. Rehydration Measurement

Rehydration experiments were carried out in a water bath at 40 °C. The weighted 5 g of dried production was put into a 250 mL beaker with 200 mL of distilled water. Samples were weighted after 0.5 h. The rehydration ratio (RR) of the samples after the drying treatment was calculated by Equation (2). All samples were carried in triplicates.
(2)$$\mathrm{RR}=\frac{m_{a}}{m_{0}}$$
where *m_a_* was the weight of rehydration samples and *m*_0_ represented the weight of dried products. 

#### 2.4.2. Color Measurement

The color of *L. decastes* samples with different drying method treatments was measured by a colorimeter (NR110, Shenzhen 3nh Technology Co., Ltd., Shenzhen, China). L, a, and b represented the three chromatic components of lightness, redness or greenness, and yellowness or blueness, respectively [18]. Then, the total color difference (ΔE) was calculated by Equation (3). All samples were performed in triplicates.
(3)$$∆E=\sqrt{{\left(L^{*}-L_{0}^{*}\right)}^{2}+{\left(a^{*}-a_{0}^{*}\right)}^{2}+{\left(b^{*}-b_{0}^{*}\right)}^{2}}$$
where *L*^*^_0_, *a*^*^_0_, *b*^*^_0_ represented the color parameters of fresh *L. decastes*, and *L*^*^, *a*^*^, *b*^*^ were the color parameters of dried products. 

#### 2.4.3. Texture Measurement

In order to examine the texture profiles of dried *L. decastes*, hardness (g), elasticity, and chewiness (g) were analyzed by a texture analyzer (TA-XT plus, Ultratech Instruments Shanghai Branch, Shanghai, China). The test speed and deformation value were 2 mm/s and 40%, respectively. The trigger force was 10 g. All samples were performed in triplicates.

#### 2.4.4. Microstructure Observation 

The microstructure of *L. decastes* with different drying technologies treatment was observed by a scanning electron microscope (Apreo 2, Thermo Fisher Technology (China) Co., Ltd., Shanghai, China) with the acceleration voltage set at 5.0 kV and observed under 5000× magnification.

### 2.5. Volatile Flavor Compound Analysis

The HS-GC-IMS (FlavourSpec^®®^, Gesellschaft für Analytische Sensor systeme mbH, Dortmund, Germany) was employed to detect the volatile flavor compounds of *L. decastes* treated by different drying methods, as previously described in the literature [16]. The dried *L. decastes* powder (0.5 g, passing through a 100-mesh sieve) and fresh product (5 g) were precisely weighed, then put it into a 20 mL headspace vial, respectively. Thereafter, the samples were incubated at 60 °C, 500 rpm for 15 min. Whereafter, 200 μL of gas from the headspace was automatically injected into the warmed injector by a syringe in a splitting mode at 85 °C. At that point, the samples were driven into a RESTEK capillary column (30 m × 0.53 mm ID) by nitrogen at a programmed flow as follows: 2 mL/min for 10 min, 10 mL/min for 10 min, 100 mL/min for 10 min, before rising to 150 mL/min for the ending. The analytes were separated at 60 °C into a column, then subsequently were ionized in the IMS ionization chamber at 45 °C. The drift gas flow was set at a constant flow of 150 mL/min. The ketones (C4–C9) were used as the external standard to calculate the retention time (RI). Volatile compounds were identified by comparing RI and the drift time of the standard in the GC-IMS library. All samples were performed in triplicates. The relative content of the volatile flavor compounds was calculated by Equation (4) [19]
(4)$$The\;relative\;content\left(\%\right)=\frac{The\;peak\;volume}{\mathrm{Total}\;\mathrm{peak}\;\mathrm{volume}}\times 100$$

### 2.6. Statistic Analysis

All of the experimental data were presented as mean ± standard deviation (SD). One-way ANOVA was performed by Duncan’s multiple range test with a significant level (*p* < 0.05). The software of IBM SPSS statistics (version 26.0, SPSS Inc., Chicago, IL, USA) was applied for significance analysis. PCA was performed by the HS-GC-IMS-assisted analysis software VOCal.

## 3. Results and Discussion

### 3.1. The Drying Time and Unit Energy Consumption of L. decastes under Different Drying Methods

Figure 1 presents the effects of the various drying techniques on the drying time and energy consumption of *L. decastes*. As shown in Figure 1a,b, VFD consumed more time (around 22 h) and the highest UEC among all drying technologies. Although the drying time of HAD was the second longest (around 14 h), it was the lowest in energy consumption. Compared with VFD and HAD, HAVD significantly (*p* < 0.05) reduced the drying time. However, it is worth noting that HAVD consumed more energy than HAD by 135% (*p* < 0.05), which was likely due to the high-energy consumption of the vacuum pump [20]. The most conventional vacuum dryers rely on conduction heat transfer from hot plates, which is slow, difficult to control, and requires a large surface area [21]. This resulted in a high-operating cost and lower drying efficiency of HAVD. Therefore, HAD exhibited not only the lowest in energy consumption, but also the lowest drying time among the three drying methods. These results agreed with the conclusions from the literature for *Agaricus bisporus* and shiitake mushrooms [22,23]. However, it has been reported that the energy consumption of the VFD process could be reduced by adjusting the working pressure and shelf temperature [24]. In the future, the drying efficiency could be further improved by promoting the drying process.

### 3.2. Physicochemical Properties of L. decastes Treated by Different Drying Techniques

#### 3.2.1. Rehydration Behaviors

The rehydration ratio (RR) is an important evaluation index of structural quality for the dried agricultural products, which indicates the degree of cellular and structural disruption in the process of drying [25]. As found in Table 1, the RR of VFD, HAVD, and HAD was 5.66, 3.12, and 2.94, respectively. The RR of the *L. decastes* dried by VFD was the highest, while the HAD sample had the lowest RR value. The previous study showed that the variations in the pore size of the product’s internal structure had a major influence on rehydration [23]. The material was exposed in the high-temperature environment for a long time during the HAD, which easily caused significant shrinkage and formed a dense structure with a reduced ability for moisture retention in the rehydration process [22]. As a result, the rehydration of HAD *L. decastes* was difficult. By contrast, the VFD sample was considered to have the best rehydration performance due to its protective effect on cells and tissue structure at a low temperature, and a highly porous structure could facilitate rehydration [26].

#### 3.2.2. Color

The color of dried mushrooms is an important parameter for food quality and consumer acceptability. It usually changes during the drying process, owing to the enzymatic browning reaction and pigment degradation [27]. As shown in Table 1, the drying technologies significantly affected the color of *L. decastes*. The VFD product had the highest value of L*, and the lowest values of a* and b*, which were significant differences from HAD and HAVD (*p* < 0.05). The similar tendency was found in ΔE—the VFD product exhibited the lowest ΔE value of 1.54, followed by HAVD, and the HAD product had the highest ΔE value. The result confirmed that VFD samples underwent less color degradation due to the vacuum and low-temperature environment that inhibited the enzymatic browning reaction and pigment degradation [28].

#### 3.2.3. Texture and Microstructure

The texture property has often been used to assess the quality of dried agriculture products [29]. The drying method generally has significant effects on the texture of dried products [30]. As shown in Table 1, the hardness of the HAD sample had reached a maximum value of 1560 g, and that of VFD products had reduced to the lowest value of 405.83 g. In addition, the elasticity and chewiness indicated the same trend. The HAD sample significantly had the highest values of elasticity and chewiness compared to the others (*p* < 0.05). This result was expected, providing that the sample with thermal treatment (HAD and HAVD) exhibited more dense structures and a severely-damaged cell structure, as observed by SEM (Figure 2a,b). However, the skeleton structure of *L. decastes* with VFD treatment was well retained due to the removal of water, which occurred by sublimation from frozen substances with the simultaneous effect of the vacuum (Figure 2c). Therefore, the VFD sample showed an excellent rehydration capacity and the lowest hardness, elasticity, and chewiness. Similar results have also been reported by other studies [11,12,31]. It is noteworthy that *Liu* et al. reported that maintaining the hardness of potatoes could reduce the loss of products and save costs during the transportation [32]. Furthermore, the dried *L. decastes* samples with higher hardness and elasticity were obtained by HAD, which was economical for packaging and convenient for transportation.

### 3.3. Volatile Flavor Compounds of L. decastes under Different Drying Technologies

The flavor components of fresh and dried *L. decastes* treated by three drying methods are shown in Figure 3a, with the drift time as abscissa and the retention time as ordinate. Each point at the right of the reactive ion peak (RIP) represents one volatile compound split from the samples, respectively. The color of the point represents the substance intensity of the volatile components. The red represents a higher concentration, while the blue indicates a lower concentration level. Compared with the fresh sample, the picture offers a clear vision of the color intensity—representing a high concentration level in the three dried samples—meaning that the drying process significantly influenced the composition of the flavor compounds of *L. decastes*. This is consistent with the report that *Lentinula edodes* via the drying process detects more volatile compounds [12].

In order to better recognize the differences in flavor components in the four samples, the visual gallery plot was built according to the signal peak intensity of each compound (Figure 3b). Each row in the gallery plot represented all of the signal peaks selected in a sample, and each column represented the signal peaks of the same volatile substance in different samples [15]. From Figure 3b, it can be seen that a total of 57 volatile compounds were successfully identified. Most of these compounds in region A of Figure 3b, such as limonene dimer, limonene, p-cymene, (Z)-6-nonenal dimer, acrolein, and benzaldehyde, were clearly abundant in fresh *L. decastes* samples, and they were extremely damaged during the drying stage because of volatilization and thermal degradation [33]. In contrast, acetone, 2-butanone, 2-butanone dimer, butanal, butanal dimer, cyclopentanone, butan-1-ol, pentanal, and so on, labelled as B, were considerably increased though the drying treatments; this can be attributed to the precursors of fresh samples which degraded or reacted with each other during the drying process [10,34]. In addition, the drying methods exerted profound effects on the volatile fingerprints of *L. decastes*. In the HAD sample, 3-hydroxy-2-butanone, (E)-2-hexenal, 1-pentanol, 1-hydroxypropan-2-one, (E)-2-heptenal, pentanal, and l-menthol exhibited obviously higher concentrations than other samples (in the yellow frame of Figure 3b). The amounts of acetic acid, pentyl acetate, 2-hydroxybenzaldehyde, and ethyl acetate were markedly higher in the HAVD sample than other samples (in the green frame of Figure 3b). Among these, octanal and heptanal were the highest in the VFD sample. Notably, compared with VFD, *L. decastes* samples dried via HAD and HAVD were detected with similar types of volatile compounds. This may be because when HAD and HAVD were both under thermal treatment, the increasing temperature of the samples promoted the Maillard reaction, which resulted in the similar flavor composition [10]. An earlier study by Zhang et al. on *Allium mongolicum* Regel also found that the flavors of VD and HAD AMR samples were relatively similar but different from the FD sample [35].

The volatile compounds in *L. decastes* subjected to different drying methods were listed in Table 2. These compounds were classified into five chemical types, including 24 aldehydes, 12 alcohols, 10 ketones, 5 alkenes, esters, and 6 other compounds. It has been reported that aldehydes, alcohols, esters, and ketones contributed significantly to the formation of the flavor profiles of foods [10,34]. To further compare the changes in flavor substances of *L. decastes* between various drying methods, the relative content of flavor components in different samples were calculated in Figure 4 according to the gallery plot. 

The aldehydes were the major chemical family of *L. decastes* after the drying process, and they contributed greatly to the flavor of mushrooms due to the low odor thresholds and intense flavor properties [36]. From Figure 4, except the other components, the relative content of aldehydes in the HAD sample was the highest with 32.87%, followed by VFD, and the HAVD sample exhibited the lowest content with 22.79%. Combined with Figure 3b, the signal intensities of nonanal, butanal, and pentanal were significantly increased by the three drying methods—this finding was consistent with the findings of others mushrooms (Luo et al., 2021). (E)-2-hexenal, (E)-2-heptenal, and pentanal appeared with the largest intensity in HAD sample. HAD and HAVD resulted in a higher level of 3-methylbutanal than VFD, which was mainly a result of the Strecker degradation [36]. Octanal (citrus orange with a green peel nuance) and heptanal (citrus odor), which were related to a desirable fruity aroma, were greatly increased via VFD. 

Alcohols were mainly formed by the oxidation of fatty acids [37]. A total of 12 alcohols were detected in fresh and dried *L. decastes* samples. Among these, butan-1-ol had a sweet balsam whiskey aroma and 3-methyl-1-butanol provided the fruity banana flavor. In agreement with the published literature, they were substantially enhanced after the drying process in this study [38]. The increase in these alcohols could be due to the high decarboxylase activity in the reaction of keto acids [39]. Ketones also resulted in a rich aroma of dried mushrooms [10]. In this study, the relative contents of ketones were significantly increased after the drying process. Similar results were mentioned in other mushrooms [40,41]. As shown in Figure 4, ketones were the highest in the HAD sample—this might have been possible because ketones were formed by the oxidation of fatty acids and the occurrence of the Maillard reaction. In addition, the thermal treatment could promote the reaction, whereas the vacuum and low-temperature treatment had reverse effects for its formation [12,42,43].

In summary, regarding the three drying methods, dried *L. decastes* via HAD was detected with a higher relative content of aldehydes, alcohols, and ketones than via HAVD and VFD. This was consistent with the research on juice [44,45]. Other than the thermal-treatment effect, this finding could be due to the destruction of the cell wall and cytoplasm via HAD, more various intracellular components released, or the involvement with the reaction during drying [16]—if considered, more flavor compounds could be produced. This agreed with the microstructure result.

### 3.4. Principal Component Analysis (PCA) of L. decastes Flavor Volatile Components

PCA is a multivariate statistical analysis technique. The complex and difficult-to-find variables in the original samples were represented by a few principal component factors, therefore, the differences from the samples could be evaluated according to the contribution rate of principal component factors of the different treatments [44,45]. The PCA of volatile substances in both fresh and different treated *L. decastes* samples was clarified in Figure 5. The accumulative contribution of the first and second principal components was 87% (PC1 was 64% and PC2 was 23%, respectively). These components indicated that the differences in dried *L. decastes* samples under different drying methods could be separated. The value of PC1 increased in the following order: FP < VFD < HAVD < HAD. The regions of each group of *L. decastes* showed that the HAVD and HAD samples were close to each other, meaning that they had similar flavors. In addition, the VFD group was located closely to FP, indicating that the flavor of the VFD samples was more similar to the FP sample than HAD and HAVD. The result agreed with a recent study in which samples of *Ganoderma lucidum*, which were dried via different treatments, had different spatial distributions of PCA [46].

## 4. Conclusions

Based on the result of the present investigation, we concluded that the drying methods had significantly affected the energy consumption, physicochemical properties, and volatile compounds of the *L. decastes*-dehydrated products. Among all the drying methods (HAD, HAVD, and VFD), VFD had a better rehydration capacity, and a lower color difference. Furthermore, VFD could also better retain the tissue structure of fresh *L. decastes*, which might be attributed to a lower operating temperature. However, it had a low efficiency and a high-power consumption. In contrast, HAD was preferred—it had a relatively lower drying time and was the most energy efficient. In addition, the higher hardness of the HAD sample was more economical for packaging and more convenient for transportation. For flavor components, HAD could effectively promote the formation of new volatile flavor compounds and provide dried *L. decastes* with a more intense aroma, while the VFD was more similar to the FP sample. Therefore, VFD would be suitable for attaining dried *L. decastes* samples with high requirements for color and appearance, while the HAD drying method was more energy-efficient, economical, and obtained a richer aroma. The above results will provide reference for selecting a drying method that maximizes the retention and the qualities of *L. decastes* in industrial productions.

## Figures and Tables

**Figure 1 foods-11-03249-f001:**
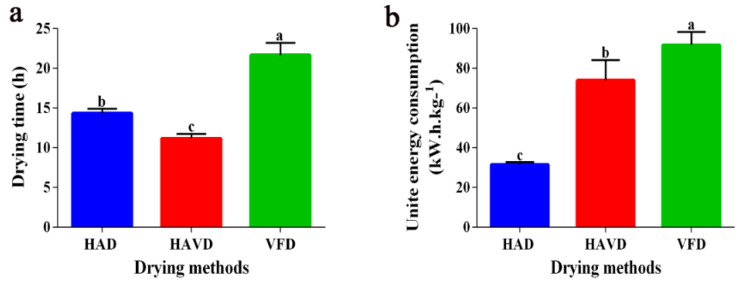
Drying time (**a**) and unit energy consumption (**b**) of *L. decastes* under different drying methods; HAD: hot-air-drying sample; HAVD: hot-air-combined-with-vacuum-drying sample; VFD: vacuum-freeze-drying sample. Different letters (a, b, c) indicate significant differences between treatments (*p* < 0.05).

**Figure 2 foods-11-03249-f002:**
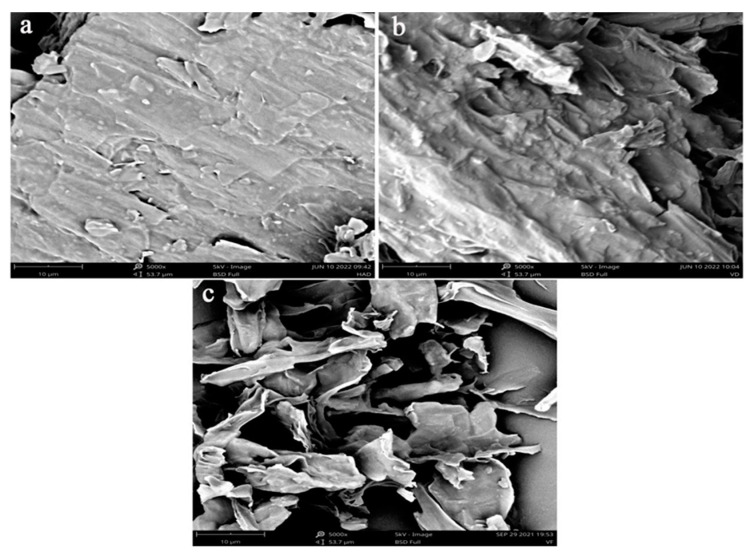
The microstructure of *L. decastes* under different drying methods: (**a**) hot-air-drying sample; (**b**) hot-air-combined-with-vacuum-drying sample; (**c**) vacuum-freeze-drying sample.

**Figure 3 foods-11-03249-f003:**
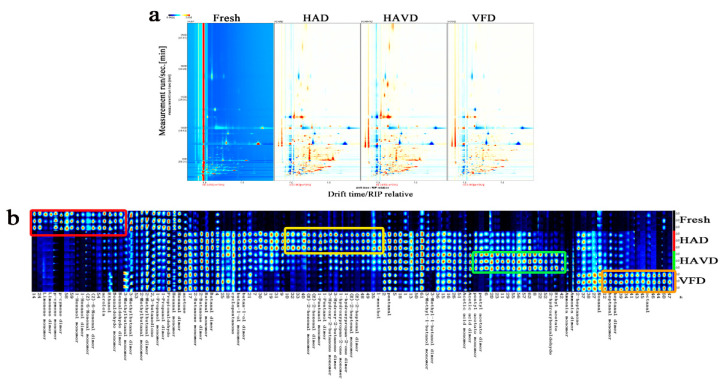
The fingerprints of the volatile flavor compounds of *L. decastes* under different drying methods, as determined by GC-IMS: (**a**) difference topographic plot, (**b**) gallery plot. HAD: hot-air-drying sample; HAVD: hot-air-combined-with-vacuum-drying sample; VFD: vacuum-freeze-drying sample.

**Figure 4 foods-11-03249-f004:**
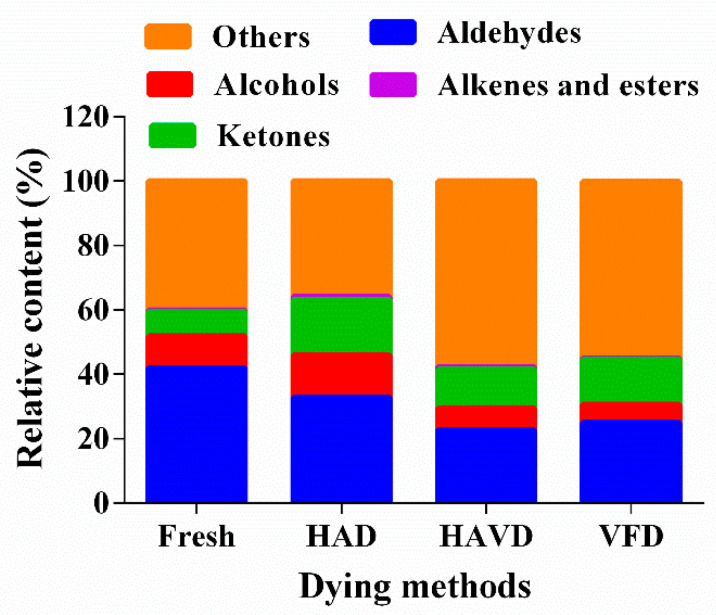
The relative content of volatile flavor compounds of *L. decastes* with different drying method treatments. HAD: hot-air-drying sample; HAVD: hot-air-combined-with-vacuum-drying sample; VFD: vacuum-freeze-drying sample.

**Figure 5 foods-11-03249-f005:**
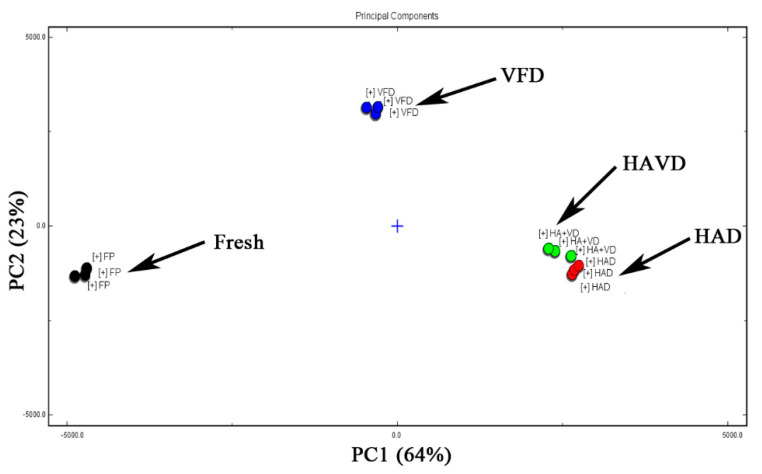
PCA analysis of flavor volatile components of *L. decastes* treated by different drying methods. HAD: hot−air−drying sample; HAVD: hot−air−combined−with−vacuum-drying sample; VFD: vacuum−freeze−drying sample.

**Table 1 foods-11-03249-t001:** Color parameters (lightness, redness or greenness, and yellowness or blueness), rehydration ratio and texture (hardness, elasticity, chewiness) of *L. decastes* treated by different drying techniques.

Drying Methods	L	a	b	ΔE	RR	Hardness (g)	Elasticity	Chewiness (g)
FP	75.44 ± 0.44 ^B^	2.89 ± 0.09 ^B^	13.93 ± 0.08 ^C^			1256.13 ± 163.56 ^B^	0.72 ± 0.02 ^B^	414.72 ± 13.52 ^B^
HAD	55.85 ± 0.05 ^C^	4.04 ± 0.06 ^A^	16.8 ± 0.72 ^B^	19.84 ± 0.16 ^A^	2.94 ± 0.03 ^C^	1560 ± 166.14 ^A^	0.91 ± 0.005 ^A^	761 ± 134.77 ^A^
HAVD	61.07 ± 0.2 ^D^	4.41 ± 0.28 ^A^	18.56 ± 0.63 ^A^	15.18 ± 0.41 ^B^	3.12 ± 0.12 ^B^	1455.28 ± 184.48 ^AB^	0.85 ± 0.04 ^A^	421 ± 66.01 ^B^
VFD	76.76 ± 0.93 ^A^	3.24 ± 0.28 ^B^	13.54 ± 0.33 ^C^	1.54 ± 0.72 ^C^	5.66 ± 0.05 ^A^	405.83 ± 26.06 ^C^	0.51 ± 0.07 ^C^	385.23 ± 33.44 ^B^

Means ± SD, *n* = 3; FP: fresh sample; HAD: hot-air-drying sample; HAVD: hot-air-combined-with-vacuum-drying sample; VFD: vacuum-freeze-drying sample; L: lightness; a: redness or greenness; b: yellowness or blueness; ΔE: color difference value; RR: rehydration ratio. Means with different superscript (A, B, C, D) in the same line are significantly different (*p* < 0.05).

**Table 2 foods-11-03249-t002:** Identification of volatile flavor compounds in *L. decastes* with different drying method treatments.

Categories	Count	Compound	CAS	Formula	MW	RI	Odor Description
Aldehydes	1	(E)-Heptenal dimer	18829-55-5	C_7_H_12_O	112.2	1325.2	Pungent; green; fatty
	2	(E)-Heptenal	18829-55-5	C_7_H_12_O	112.2	1326.1	
	3	(E)-2-Hexenal dimer	6728-26-3	C_6_H_10_O	98.1	1224.9	Powerful green vegetable-like
	4	(E)-2-Hexenal	6728-26-3	C_6_H_10_O	98.1	1225.8	
	5	(Z)-6-Nonenal dimer	2277-19-2	C_9_H_16_O	140.2	1411	Very diffusive melon-like odor, slightly metallic
	6	(Z)-6-Nonenal	2277-19-2	C_9_H_16_O	140.2	1409.2	
	7	2-Hydroxybenzaldehyde	90-02-8	C_7_H_6_O_2_	122.1	1678.4	Medical spicy cinnamon; Wintergreen cooling
	8	2-Methylbutanal dimer	96-17-3	C_5_H_10_O	86.1	919.7	Chocolate; nutty
	9	2-Methylbutanal	96-17-3	C_5_H_10_O	86.1	916.6	
	10	3-Methylbutanal dimer	590-86-3	C_5_H_10_O	86.1	928.5	Ethereal aldehydic; chocolate; peach
	11	3-Methylbutanal	590-86-3	C_5_H_10_O	86.1	931.3	
	12	Acrolein	107-02-8	C_3_H_4_O	56.1	865.6	Almond cherry
	13	Benzaldehyde dimer	100-52-7	C_7_H_6_O	106.1	1543.4	Almond fruity; powdery nutty cherry; maraschino cherry
	14	Benzaldehyde	100-52-7	C_7_H_6_O	106.1	1542.1	
	15	Butanal dimer	123-72-8	C_4_H_8_O	72.1	886.6	Pungent cocoa; musty green malty bready
	16	Butanal	123-72-8	C_4_H_8_O	72.1	885.5	Musty green malty bready
	17	Heptanal dimer	111-71-7	C_7_H_14_O	114.2	1191.3	Fresh, green, citrus odor
	18	Heptanal	111-71-7	C_7_H_14_O	114.2	1191.3	
	19	Hexanal dimer	66-25-1	C_6_H_12_O	100.2	1095	Fruity and clean with a woody nuance
	20	Hexanal	66-25-1	C_6_H_12_O	100.2	1095.9	
	21	Nonanal	124-19-6	C_9_H_18_O	142.2	1399.4	Citrus; fresh, green lemon peel
	22	Octanal	124-13-0	C_8_H_16_O	128.2	1294.7	Citrus orange with a green peel nuance
	23	Pentanal	110-62-3	C_5_H_10_O	86.1	998.5	Fruity with berry nuances
	24	Propionaldehyde	123-38-6	C_3_H_6_O	58.1	821	Wine-like; chocolate
Alcohols	25	1-Hexanol dimer	111-27-3	C_6_H_14_O	102.2	1362.4	Fruity and alcoholic
	26	1-Hexanol	111-27-3	C_6_H_14_O	102.2	1363.3	
	27	1-Pentanol dimer	71-41-0	C_5_H_12_O	88.1	1256.7	Pungent; yeasty; winey
	28	1-Pentanol	71-41-0	C_5_H_12_O	88.1	1258	
	29	1-Propanol dimer	71-23-8	C_3_H_8_O	60.1	1047.2	With a slightly sweet fruity nuance of apple and pear
	30	1-Propanol	71-23-8	C_3_H_8_O	60.1	1049	
	31	3-Methyl-1-butanol dimer	123-51-3	C_5_H_12_O	88.1	1212.9	Whiskey; fruity banana
	32	3-Methyl-1-butanol	123-51-3	C_5_H_12_O	88.1	1213.3	
	33	Butan-1-ol dimer	71-36-3	C_4_H_10_O	74.1	1150.2	Fusel oil; sweet balsam whiskey
	34	Butan-1-ol	71-36-3	C_4_H_10_O	74.1	1150.2	
	35	Ethanol	64-17-5	C_2_H_6_O	46.1	942.9	Alcoholic
	36	L-menthol	2216-51-5	C_10_H_20_O	156.3	1661.5	Minty
Ketones	37	1-Hydroxypropan-2-one dimer	116-09-6	C_3_H_6_O_2_	74.1	1304.2	Pungent sweet caramellic ethereal
	38	1-Hydroxypropan-2-one	116-09-6	C_3_H_6_O_2_	74.1	1305.1	
	39	2,3-Butanedione	431-03-8	C_4_H_6_O_2_	86.1	990.3	Sweet; creamy; buttery
	40	2-Butanone dimer	78-93-3	C_4_H_8_O	72.1	909.2	Diffusive and slightly fruity with a camphoraceous nuance
	41	2-Butanone	78-93-3	C_4_H_8_O	72.1	910.3	
	42	2-Heptanone	110-43-0	C_7_H_14_O	114.2	1188.3	Cheese; fruity; green banana
Ketones	43	3-Hydroxy-2-butanone dimer	513-86-0	C_4_H_8_O_2_	88.1	1289.7	Sweet; buttery; creamy; dairy; milky; fatty
	44	3-Hydroxy-2-butanone	513-86-0	C_4_H_8_O_2_	88.1	1288.8	
	45	Acetone	67-64-1	C_3_H_6_O	58.1	839.8	Apple; pear
	46	Cyclopentanone	120-92-3	C_5_H_8_O	84.1	1142.6	Minty
Alkenes and esters	47	Ethyl acetate	141-78-6	C_4_H_8_O_2_	88.1	893.4	Grape and rum-like
	48	Limonene dimer	138-86-3	C_10_H_16_	136.2	1199	Citrus orange fresh sweet
	49	Limonene	138-86-3	C_10_H_16_	136.2	1197.4	
	50	Pentyl acetate dimer	628-63-7	C_7_H_14_O_2_	130.2	1180.7	Fruity of banana and pear
	51	Pentyl acetate	628-63-7	C_7_H_14_O_2_	130.2	1181.3	
Others	52	Acetic acid dimer	64-19-7	C_2_H_4_O_2_	60.1	1494.4	Sharp pungent sour vinegar
	53	Acetic acid	64-19-7	C_2_H_4_O_2_	60.1	1495.3	
	54	Ammonia dimer	7664-41-7	H_3_N	17	1258.3	Ammoniacal odor
	55	Ammonia	7664-41-7	H_3_N	17	1245.1	
	56	P-cymene dimer	99-87-6	C_10_H_14_	134.2	1261.7	Woody and terpy-like with an oxidized citrus lemon
	57	P-cymene	99-87-6	C_10_H_14_	134.2	1259.4	

Notes: MW: molecular mass. RI: retention index. The odor descriptions of the volatile flavor compounds were referenced by the TGSC Information System.

## Data Availability

Data is contained within the article.

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
