# Peer review of "Effects of Drying Methods on the Physicochemical Aspects and Volatile Compounds of Lyophyllum decastes"

_foods, 2022, doi:10.3390/foods11203249_

Round 1

Reviewer 1 Report

The work evaluated different drying methods to produce L. decastes. 

It is an interesting study with the intention to give options to industries to industrialise L. decastes with the best physicochemical properties and in the faster and economical way possible. 

I would recommend to publish with minor modifications. 

Abstract

Why do you say HAD has best flavor quality? Isn’t VFD that one retaining more volatile? It is a bit confusion along the text which method between HAD and VFD is the one retaining volatile compounds.

Introduction

You could provide a bit more about information and data of consumption/production of  Lyophyllum decastes, so explaining why it is relevance.

 2.1. Materials

Just briefly explain how you measured the moisture.

 2.2. Drying treatment

Explain why you dried until 10% moisture.

 2.2.1. Hot air drying

The drying parameters were primary temperature at 42C for 3 h, and then rose to 55C for how long?? If it is not like that, please try to describe clearer the conditions.

2.2.2 HAVD

Same comments as 2.2.1.

 2.2.3. VFD

 Write the brand of the refrigerator. And also the time of this treatment.

2.4.3. Texture measurement

 Which parameters did you measure?

Results

3.1. The drying time and unit energy consumption of L. decastes under different drying methods

 Regarding the drying time, as you said that you had already calculated it previous experiments, there is some missing information on how you you reach the 10% of moisture. If you are going to show the drying time as a result, a description on how you calculated the time needed to dry up to 10 % needs to be done. E.g. did you use a moisture analyser during the drying process? Otherwise, you could remove this part of the results and just mention the time in the part of material and methods as suggested in the above sections.

 3.2.3 Texture and microstructure

this phrase should be in Material and methods: “In order to 218 examine the texture profiles of dried L. decastes, hardness, elasticity, and chewiness were 219 analyzed”  with units.

 Table 1: Add in footnote definition of RR.

 3.3 Volatile flavor compounds of L. decastes under different drying technologies 248

Line 264-266: you said that the concentration of aldehydes in higher in the fresh and VFD but then in Line 291-292: you said that HAD had the highest aldehydes concentration according to the relative content of flavor. Please, check this.

Also, how you calculated the relative content of flavour?

 Conclusions

Why do you say that VFD could have better retain the flavor substance? But then you said HAD was the best flavor quality.

 Also, you suggested HAD as best from economical point of view and VFD for physicochemical characteristics. However, the idea is to give a solution to industries to produce L. decastes in a effective way considering both the quality of the final product and the economical point of view. I would suggest that you could consider highlighting VDF as the suitable for physicochemical characterisation and the possibility to reduce the time using other variables like heating the plate of chamber that reduces the time. ( https://doi.org/10.3390/foods10112756 ). I think in that way the work would be more rounded. 

References

Line 425: add coma after 2021.

Double check in commas and spaces in references.

Author Response

Dear Reviewer,

  At first, thank you for your advice, which is very helpful for us to correct the errors and improve the quality of the paper.  According to your suggestions, we have made modifications one by one. Please see the attachment for details.

Reviewer 2 Report

Line 48: double check format for citation

Line 148: How the author specifically measures 200uL of headspace gas to be injected into GC-IMS? is there any specific method that the author follow for the extraction of volatile compounds such as dynamic headspace or SPME?I found that the extraction was quite crude if totally depending on auto-sampling.

In general, the discussions are quite general and not emphasize on specific /details results and discussions. The results provided contained many data but the discussion did not focus and highlight the importance. For example, the results on volatile compounds, few compounds were identified to be different among each drying methods. was that due to the amount or the presence of that particular compounds? No specific reason how the compounds are produced or the pathway it can be produced were only briefly mentioned. Instead, only general statement was given (line 297). 

The authors has a very good analysis of the PCA using topographic and gallery plot so the information must be fully utilized and discussed. 

Table 2: instead of using CAS no, I think it would be better to include the odor description for each compounds.

Author Response

(The authors gave the same response as above.)
